# Plasma Insulin-like Growth Factor-Binding Protein-2 of Critically Ill Patients Is Related to Disease Severity and Survival

**DOI:** 10.3390/biomedicines11123285

**Published:** 2023-12-12

**Authors:** Patricia Mester, Ulrich Räth, Luisa Popp, Stephan Schmid, Martina Müller, Christa Buechler, Vlad Pavel

**Affiliations:** Department of Internal Medicine I, Gastroenterology, Hepatology, Endocrinology, Rheumatology, and Infectious Diseases, University Hospital Regensburg, 93053 Regensburg, Germany; patricia.mester@klinik.uni-regensburg.de (P.M.); ulrich.raeth@stud.uni-regensburg.de (U.R.); luisa.popp@stud.uni-regensburg.de (L.P.); stephan.schmid@klinik.uni-regensburg.de (S.S.); martina.mueller-schilling@klinik.uni-regensburg.de (M.M.); vlad.pavel@klinik.uni-regensburg.de (V.P.)

**Keywords:** sepsis, SIRS, COVID-19, dialysis, procalcitonin, survival

## Abstract

Insulin-like growth factor-binding protein (IGFBP)-2 regulates the bioactivity of the anabolic hormone’s insulin-like growth factors, which are decreased in sepsis and contribute to the catabolic status of severely ill patients. The circulating levels of IGFBP-2 in critical illness have been rarely studied; therefore, we evaluated IGFBP-2 plasma levels in patients with systemic inflammatory response syndrome (SIRS) or sepsis as well as healthy controls. Our analysis of 157 SIRS/sepsis patients revealed higher plasma IGFBP-2 levels compared to 22 healthy controls. Plasma IGFBP-2 levels correlated positively with procalcitonin but not with C-reactive protein, interleukin-6, or the leukocyte count. Septic shock patients exhibited higher IGFBP-2 levels than those with SIRS. Bacterial or SARS-CoV-2 infection did not influence plasma IGFBP-2 levels. There was no difference in the IGFBP-2 levels between ventilated and non-ventilated SIRS/sepsis patients, and vasopressor therapy did not alter these levels. Dialysis patients had elevated plasma IGFBP-2 levels. Survivors had lower plasma IGFBP-2 levels than non-survivors. In conclusion, our study indicates that plasma IGFBP-2 levels are associated with disease severity, renal failure, and mortality in SIRS/sepsis patients.

## 1. Introduction

Sepsis results from a dysregulated immune response to viral, fungal, or bacterial infections [1,2]. The SARS-CoV-2 infection has recently emerged as an important cause of sepsis, with a significant number of hospitalized COVID-19 patients developing sepsis due to the virus [3]. In relatively many sepsis patients, neither bacterial nor viral infections can be detected [3,4,5].

Sepsis is characterized by an initial phase of excessive inflammation, followed by a state of immunosuppression. Strategies targeting the cytokine storm in early disease have not been successful. While most sepsis studies have focused on inflammatory processes, anti-inflammatory pathways have received significantly less attention [1,6,7].

Sepsis is a hypercatabolic state associated with a loss of muscle mass. Convincing evidence suggests the role of low levels of the anabolic hormones insulin-like growth factors (IGFs) in the catabolic status of sepsis [8,9]. IGF-regulated pathways are evolutionarily conserved and regulate the growth of almost every organ in the body [10].

Insulin-like growth factor-binding protein (IGFBP)-2 regulates the bioactivity of IGFs. The IGFBP-2 transgenic mouse revealed that IGFBP-2 negatively affects postnatal growth; this effect was attributed to the reduced local bioavailability of IGFs [11].

Furthermore, IGFBP-2 exerts diverse functions, independent of IGFs. This protein can translocate into the cell, bind proteins like p21, and even enter the nucleus to regulate gene expression [12,13].

IGFBP-2 plays a significant role as an oncogene in various tumors, promoting processes such as cell proliferation, invasion, and migration [14]. The immune environment within tumors is vital for tumorigenesis, and IGFBP-2 has displayed immunosuppressive effects in glioblastomas [15]. In pancreatic ductal adenocarcinoma cells, IGFBP-2 stimulates the expression of IL-10. This anti-inflammatory cytokine then drives the polarization of tumor-associated macrophages to an immunosuppressive M2-like phenotype [16]. Consistent with its oncogenic effects, high IGFBP-2 expression in tumor tissues correlates with poorer prognosis [17]. IGFBP-2 holds potential as a therapeutic target in cancer, and circulating levels might be useful as diagnostic and prognostic biomarkers [14].

A dysfunctional immune system characterizes sepsis [2], yet the role of IGFBP-2 in critical illness remains understudied. Peripheral immune cells from patients with severe sepsis express higher levels of IGFBP-2 than those from healthy controls, suggesting a potential involvement of IGFBP-2 in immune cell dysfunction [18]. To our knowledge, only two studies have analyzed circulating levels of IGFBP-2 in critically ill patients. One found that critically ill patients had levels of IGFBP-2 above the normal range upon hospital admission [19]. The other reported that patients with severe illness had double the systemic levels of IGFBP-2 compared to healthy controls [20]. Hemolytic uremic syndrome, a serious complication arising from infection with enterohemorrhagic *Escherichia coli* [21], found that patients with severe cases displayed higher serum levels of IGFBP-2 than healthy controls. Furthermore, the serum levels of IGFBP-2 correlated positively with markers indicating the disease severity of patients with hemolytic uremic syndrome [22].

Sepsis-associated acute kidney injury is a frequent complication in critically ill patients [23]. Studies on male rats with acute kidney injury showed high IGFBP-2 protein expression in their renal tissues [24]. Additionally, patients with conditions like lupus nephritis, diabetic kidney disease, and chronic kidney diseases have been shown to exhibit elevated serum IGFBP-2 levels [12,25].

While obesity is characterized by low-grade chronic inflammation [26], evidence consistently shows a decline in circulating IGFBP-2 levels among obese individuals. Reduced IGFBP-2 levels are associated with adiposity, metabolic syndrome, and type 2 diabetes. Overexpressing IGFBP-2 has been found to guard against obesity and insulin resistance by inhibiting adipogenesis and enhancing insulin sensitivity [13].

It should be noted that systemic IGFBP-2 levels predicted the mortality of healthy volunteers enrolled in the Baltimore Longitudinal Study of Aging [27]. The Health, Aging, and Body Composition study also observed an association between increased IGFBP-2 and all-cause mortality [28]. Plasma IGFBP-2 was a strong predictor for mortality in patients with heart failure [29]. In patients with acute coronary syndrome, higher plasma IGFBP-2 was associated with the development of adverse cardiovascular events [30].

Our study aimed to evaluate the associations between plasma IGFBP-2 levels and disease severity and outcome in SIRS/sepsis patients. We measured IGFBP-2 in the plasma of 157 critically ill patients with systemic inflammatory syndrome (SIRS), sepsis, or septic shock.

## 2. Materials and Methods

### 2.1. Study Cohort

Between August 2018 and January 2023, we collected plasma from 157 patients at the University Hospital of Regensburg. This included 39 patients with systemic inflammatory response syndrome (SIRS), 40 with sepsis, and 78 with septic shock. Within this cohort, 24 were infected with SARS-CoV-2; their plasma was collected from October 2020 to January 2023. We used the Sepsis-3 criteria for patient categorization [31]. Those not developing sepsis during their intensive care unit stay, but meeting the SIRS criteria, were classified as having SIRS [32]. We excluded patients with multi-resistant infections, viral hepatitis, or HIV.

Most of the plasma samples analyzed in this study were used in our earlier studies [33,34]. Our control group consisted of 22 volunteers, 11 males and 11 females, comprising hospital employees and medical students.

### 2.2. IGFBP-2 and Interleukin-6 ELISA

Blood samples were collected 12 to 24 h after patients were admitted to the intensive care unit. We used EDTA as an anticoagulant and prepared the plasma. The human IGFBP-2 DuoSet ELISA (R&D Systems; Wiesbaden, Nordenstadt, Germany) was utilized, as recommended by the manufacturer. Plasma samples were diluted 1:100 and the IGFBP-2 levels were measured in duplicate. The human interleukin-6 DuoSet ELISA kit from R&D Systems was used to measure interleukin-6, and the plasma was two-fold diluted for this analysis. For calculations, we used the mean values.

### 2.3. Statistical Analysis

Boxplots display the minimum, maximum, first and third quartiles, and the median. Outliers are depicted as circles or asterisks. Tables list the median, minimum, and maximum values. Data were analyzed using the Mann–Whitney U test, Chi-Square test, Kruskal–Wallis test, receiver operating characteristic curve, and Spearman’s correlation with the IBM SPSS Statistics 26.0 program. A *p*-value of <0.05 was considered significant.

## 3. Results

### 3.1. IGFBP-2 in Plasma of Controls, SIRS/Sepsis Patients, and SIRS/Sepsis Patients with Liver Cirrhosis

The plasma IGFBP-2 concentrations of 157 patients with SIRS/sepsis were measured. Details of this cohort are given in Table 1. The 22 controls were matched for sex and age. As expected [35], IL-6 in the plasma of the controls was significantly lower in comparison to the patients (Table 1).

The average plasma IGFBP-2 concentration in the 157 patients with SIRS/sepsis was 370 (19–1307) ng/mL, significantly higher than the 22 controls, who had an average concentration of 119 (35–389) ng/mL (*p* < 0.001; Figure 1a). The receiver operating characteristic (ROC) curve for SIRS/sepsis diagnosis had an area under the ROC curve (AUROC) of 0.900 (Figure 1b). The standard error was 0.029, significance was *p* < 0.001, and 95% confidence interval (CI) was 0.843–0.958. A plasma level of 232 ng/mL IGFBP-2 discriminated SIRS/sepsis patients from controls, with 77% sensitivity and 91% specificity.

Both men and women in the control and SIRS/sepsis groups had comparable IGFBP-2 levels (*p* = 0.056 and 0.903, respectively). There was no correlation between IGFBP-2 plasma levels and age; the Spearman correlation coefficient was r = 0.363 (*p* = 0.097) for controls and r = 0.032 (*p* = 0.690) for the SIRS/sepsis group. Patients’ IGFBP-2 levels were not related to their body mass indexes (BMI) (r = −0.031, *p* = 0.708).

The patients were categorized as SIRS, sepsis, and septic shock [31] (Table 1). IGFBP-2 in plasma was higher in septic shock compared to SIRS (*p* = 0.018; Figure 1c). The AUROC for septic shock diagnosis was 0.617, *p* = 0.012 and 95% CI = 0.529–0.706. Age (*p* = 0.168), C-reactive protein (CRP; *p* = 0.143), procalcitonin (*p* = 0.186), and interleukin-6 (IL-6; *p* = 0.051) levels did not differ between SIRS, sepsis, and septic shock.

### 3.2. IGFBP-2 in Plasma of SIRS/Sepsis Patients Stratified for Underlying Diseases and Infectious Diseases

In our cohort, 30 patients had liver cirrhosis, 33 developed SIRS/sepsis from pancreatitis, and 10 had sepsis due to cholangitis. There was no significant difference in plasma IGFBP-2 levels across these patient groups (Figure 2a). While IGFBP-2 is being considered as a potential tumor marker, the 14 patients in our SIRS/sepsis group with various cancers had IGFBP-2 levels similar to those without tumors (*p* = 0.796). This suggests IGFBP-2 might not be a reliable tumor marker in septic patients.

Common infections that lead to sepsis were pulmonary (53 patients) and urinary tract infections (15 patients). The plasma IGFBP-2 levels of these groups were similar (*p* = 0.211; Figure 2b).

In our study, the 24 septic COVID-19 patients exhibited plasma IGFBP-2 levels comparable to the SIRS/sepsis patients without a SARS-CoV-2 infection (*p* = 0.606; Figure 3a). Although SARS-CoV-2-infected patients had lower procalcitonin levels (*p* = 0.020) than non-infected patients, their IL-6, CRP, and leukocyte counts remained consistent (Table 1).

Even though some studies indicate increased procalcitonin in severe viral infections [36], it is widely accepted that procalcitonin can help differentiate between viral and bacterial infections or superinfections [37,38]. Based on our findings, low procalcitonin was not appropriate to discriminate patients with and without SARS-CoV-2 infection (AUROC 0.352; *p* = 0.021; 95% CI 0.240–0.463).

The plasma IGFBP-2 levels of COVID-19 patients were higher in contrast to the healthy controls (*p* < 0.001; Figure 3b) (AUROC 0.879; *p* < 0.001; 95% CI 0.773–0.985).

### 3.3. Plasma IGFBP-2 in Relation to Vasopressor Therapy and Interventions

We examined the associations of plasma IGFBP-2 levels with the need for dialysis, ventilation, or vasopressor treatment. Patients requiring dialysis had elevated plasma IGFBP-2 levels and also showed an increase in procalcitonin (*p* = 0.003) and IL-6 (*p* = 0.022) compared to those not on dialysis. The AUROC for dialysis was 0.620, *p* = 0.015 and 95% CI = 0.528–0.711. There were no significant differences in age (*p* = 0.731), CRP (*p* = 0.705), or leukocyte count (*p* = 0.600) between the two groups. The plasma IGFBP-2 concentrations were not associated with the need for ventilation or vasopressor therapy (Table 2).

Of the patients with septic shock, 47 out of 78 required dialysis. This number was significantly higher compared to the sepsis group (4 out of 40 patients) and the SIRS group (1 out of 39 patients; *p* < 0.001). Elevated plasma IGFBP-2 levels in patients with septic shock (Figure 1c) may indicate either disease severity or renal dysfunction. However, comparing septic shock patients with and without dialysis showed similar plasma IGFBP-2 levels (*p* = 0.377). This suggests that disease severity, rather than kidney dysfunction, is associated with the rise in plasma IGFBP-2 levels.

### 3.4. Plasma IGFBP-2 in Relation to Inflammatory Markers

In the SIRS/sepsis cohort, plasma IGFBP-2 positively correlated with procalcitonin. However, no associations were found with leukocyte count, CRP, or IL-6 (Table 3). While IL-6 and ferritin are proposed markers for COVID-19 [39,40], they did not correlate with the plasma IGFBP-2 levels in our COVID-19 subgroup (Table 3).

### 3.5. Plasma IGFBP-2 Levels in Gram-Negative and Gram-Positive Infection

The plasma IGFBP-2 levels in patients without bacterial infection were similar to those with Gram-negative (61 patients), Gram-positive (22 patients), or both types of bacterial infections (20 patients) (*p* = 0.064; Figure 3a). No significant difference was found when comparing patients with and without bacterial infections (*p* = 0.081). Infection with either Gram-negative (*p* = 0.548) or Gram-positive (*p* = 0.747) bacteria did not notably alter plasma IGFBP-2 levels in comparison to all patients not infected by Gram-positive or Gram-negative bacteria, respectively.

### 3.6. Plasma IGFBP-2 Levels and Survival

In our cohort, plasma IGFBP-2 levels were higher in the 37 patients who died compared to survivors (*p* = 0.028; Figure 4b). The AUROC for non-survival diagnosis was 0.618, *p* = 0.031 and 95% CI = 0.515–0.720. Among the nine COVID-19 non-survivors, no significant difference in IGFBP-2 levels was observed compared to the survivors, suggesting that IGFBP-2 might not reliably predict mortality in COVID-19 sepsis (*p* = 0.222). For those requiring dialysis, plasma IGFBP-2 levels were consistent between the 31 non-survivors and survivors (*p* = 0.904). Similarly, the six non-survivors who did not need dialysis had comparable IGFBP-2 levels to the survivors (*p* = 0.114).

## 4. Discussion

This study revealed that plasma IGFBP-2 levels in SIRS/sepsis patients are over three times higher than in healthy controls. Plasma IGFBP-2 correlated positively with the need for dialysis, disease severity, and mortality.

Reliable laboratory biomarkers for the early diagnosis of sepsis are still lacking [41]. Sepsis can be clinically suspected using the Sepsis-3 criteria [31], but there are conditions such as adrenal insufficiency, diabetic ketoacidosis, or hypovolemia that can mimic the symptomatology of sepsis. In these situations, biomarkers like IGFBP-2 or mid-regional pro-adrenomedullin levels (MR-proADM) [42] could confirm sepsis.

The binding of IGFBP-2 to IGFs diminishes the bioavailability of these growth factors [43]. Human sepsis reduces systemic IGF levels, which can lead to inflammation-induced muscle atrophy and cachexia [44]. The further blockage of IGF activity by elevated IGFBP-2 levels in SIRS/sepsis—as shown in our study—might intensify these catabolic processes, leading to sarcopenia and cachexia.

Blocking IGFBP-2 activity may improve the catabolic status in sepsis and may also boost the immune response because of the immunosuppressive effects of IGFBP-2 [15,16]. Immune suppression accounts for most sepsis-related deaths [2] and IGFBP-2 may become a therapeutic target.

SARS-CoV-2 infection can result in sepsis [3,45], and nearly 90% of our COVID-19 patients experienced septic shock. Plasma IGFBP-2 levels in COVID-19 patients were higher compared to healthy controls but were similar to those with SIRS/sepsis from other causes. This suggests that elevated plasma IGFBP-2 marks critical illness, not specifically SARS-CoV-2 infection. Pneumonia, especially prevalent in COVID-19 patients [46], did not correlate with changes in plasma IGFBP-2.

Bacterial bloodstream infection did not alter plasma IGFBP-2 levels, which matched those in patients without detectable bloodstream infections. Urosepsis, commonly caused by enterobacteria and Gram-positive bacteria [47], resulted in IGFBP-2 levels similar to other SIRS/sepsis patients. Sheep injected with lipopolysaccharide maintained standard IGFBP-2 levels for the 9 h observation [48]. When endotoxin was given to humans, IGFBP-2 levels rose by 50% above baseline 3 h post-administration, remaining elevated for the next 2 h [49]. Our findings did not show elevated plasma IGFBP-2 levels in bacterially infected patients, suggesting that initial effects might disappear during follow-up.

Plasma IGFBP-2 did not correlate with inflammatory markers like the leukocyte count or CRP. While IL-6 and ferritin are indicators of COVID-19 severity [50], they did not correlate with plasma IGFBP-2. Plasma IGFBP-2 positively correlated with procalcitonin levels in the SIRS/sepsis cohort. COVID-19 patients had lower procalcitonin levels in comparison to the whole SIRS/sepsis patient group. Notably, low procalcitonin was not an independent marker for suspected COVID-19 in our cohort.

Septic shock patients had higher plasma IGFBP-2 than SIRS patients, suggesting the association of IGFBP-2 with disease severity. Since the CRP and procalcitonin levels remained consistent across patients with SIRS, sepsis, or septic shock, IGFBP-2 may be a superior marker. However, the AUROC of IGFBP-2 for the diagnosis of septic shock was 0.617, indicating poor discriminative ability.

Biomarkers for sepsis and septic shock diagnosis have been described previously [51,52]; for example, MR-proADM levels were high in sepsis and highest in septic shock. The AUROC of MR-proADM for sepsis diagnosis was 0.88 (sensitivity 78.5%, specificity 85%) [42]. The AUROC of IGFBP-2 for sepsis diagnosis described in this study was 0.90 (sensitivity 77%, specificity 91%). Since our present results revealed that IGFBP-2 was also increased in patients with septic shock, measuring both biomarkers could provide more diagnostic and prognostic accuracy in these situations. Larger studies are needed to prove this hypothesis.

Patients requiring dialysis exhibited higher plasma IGFBP-2 levels than those without dialysis. Previous reports have also indicated elevated IGFBP-2 levels in patients with kidney diseases [12]. Dialysis was more frequently required by patients with septic shock compared to those with SIRS or sepsis. Acute kidney injury is common in severe illnesses, with 16% to 37% of sepsis patients developing this complication. The prevalence rises to 64% to 78% in patients with septic shock [53]. However, comparing septic shock patients with and without dialysis showed similar plasma IGFBP-2 levels, and the role of plasma IGFBP-2 as a marker of kidney injury in sepsis needs to be studied further.

A recent study by Lacquaniti et al. investigated the potential role of the product of urinary tissue inhibitor metal proteinase 2 (TIMP2) and insulin-like growth factor-binding protein-7 (IGFBP-7). These metabolites can be assessed by a commercial immunoassay (Nephrocheck^®^ test) [54]. The authors found that urinary TIMP2 × IGFBP-2 is an early predictor of acute kidney injury and increased precociously in acute septic kidney injury after cardiac surgery [42]. The sensitivity of this test was 83.9% and the specificity was 73.8% [42]. It should be noted that lower sensitivities of 61.54% for the detection of acute kidney injury in sepsis were reported [55]. A combination of other biomarkers with the Nephrocheck^®^ test could increase the specificity and sensitivity achievable when diagnosing acute septic kidney failure. Future research will show whether plasma IGFBP-2 could be a complementary marker to other tools like the Nephrocheck^®^ test in septic patients.

In our cohort, 60% of the septic shock patients required dialysis, compared to 10% of the sepsis patients and 2.6% of the SIRS patients. These figures align with previously published data [53]. When stratifying septic shock patients based on dialysis requirement, plasma IGFBP-2 levels were similar for both groups. This suggests that elevated plasma IGFBP-2 levels in patients with septic shock correlate with disease severity, rather than specific kidney failure.

However, since the current literature indicates that IGFBP-2 may play a role in kidney diseases and targeting, to find an early marker for renal insufficiency in septic patients, IGFBP-2 should be further investigated in larger cohorts of septic patients with renal failure.

The plasma IGFBP-2 levels of patients surviving SIRS/sepsis were lower in comparison to non-survivors. Associations of circulating IGFBP-2 with mortality have been documented in the general population [27,28]. The highest quartile of plasma IGFBP-2 correlated with cardiovascular mortality in both acute and chronic heart failure patients [29], and serum IGFBP-2 levels exceeding 275 ng/mL predicted mortality in patients with severe aortic stenosis [56]. Elevated circulating IGFBP-2 levels have also been linked to non-survival in cancer patients [57,58]. These findings suggest that higher circulating IGFBP-2 levels correlate with increased mortality across various diseases and the general population. However, to our knowledge, such an association has not been reported for sepsis patients so far. Based on the relatively small increase in plasma IGFBP-2 levels in non-survivors, plasma IGFBP-2 is not recommended as a clinical biomarker for the prediction of mortality.

In the control and the SIRS/sepsis cohort, plasma IGFBP-2 levels did not differ between sexes. Moreover, there was no correlation between the plasma IGFBP-2 levels and age or BMI. Male sex and higher age are risk factors for non-survival [59,60,61] but did not differ in our cohort between survivors and non-survivors. The BMIs of our survivors and non-survivors were similar, and whether obesity contributes to survival in critical illness remains controversial [62]. Our study cohort may have been too small to identify differences between survivors and non-survivors. However, it must be noted that studies examining the relationship between sex and mortality in sepsis are inconclusive, and the higher, unchanged, and lower mortality of males in comparison to females was reported [60]. In accordance with the literature describing a higher prevalence of sepsis in males [60], there were twice as many men as women in our cohort.

In our SIRS/sepsis cohort, liver cirrhosis, cholangitis, and pancreatitis were prevalent underlying conditions. Serum IGFBP-2 levels were notably increased in patients with chronic hepatitis C, especially in advanced fibrosis and cirrhosis [63]. In contrast to the type 2 diabetes patients used as controls, chronic pancreatitis patients exhibited nearly 10-fold higher serum IGFBP-2 levels [64]. To our knowledge, the IGFBP-2 levels in patients with cholangitis have not been studied. In our cohort, the IGFBP-2 levels were comparable among patients with liver cirrhosis, cholangitis, and pancreatitis. This indicates that elevated IGFBP-2 might serve as a marker for critical illness, rather than being specific to individual disease entities.

## 5. Conclusions

The current study reveals that SIRS/sepsis patients exhibit higher plasma IGFBP-2 levels than healthy controls. Notably, COVID-19 patients also had elevated IGFBP-2 plasma levels compared to controls. Furthermore, in the SIRS/sepsis cohort, elevated plasma IGFBP-2 was observed in septic shock patients, those requiring dialysis, and non-survivors. This suggests the potential of IGFBP-2 as a marker for disease severity and its broader clinical applicability in critically ill patients.

## Figures and Tables

**Figure 1 biomedicines-11-03285-f001:**
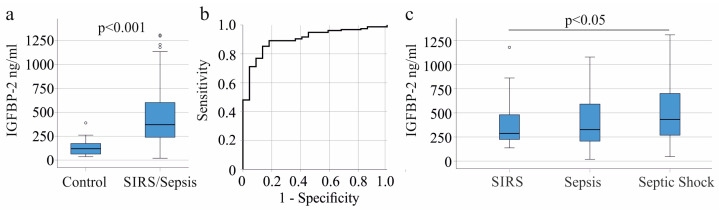
IGFBP-2 in plasma of controls and SIRS/sepsis patients. (**a**) Plasma IGFBP-2 levels of the 22 controls and the 157 SIRS/sepsis patients; (**b**) receiver operating characteristic curve of IGFBP-2 for SIRS/sepsis diagnosis; (**c**) plasma IGFBP-2 levels of patients with SIRS, sepsis, and septic shock.

**Figure 2 biomedicines-11-03285-f002:**
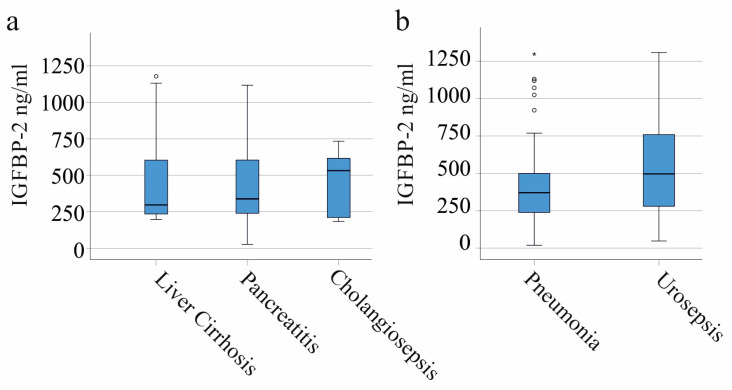
IGFBP-2 in plasma of patients with SIRS/sepsis stratified for underlying diseases and causes of SIRS/sepsis. (**a**) Plasma IGFBP-2 levels of SIRS/sepsis patients with liver cirrhosis, pancreatitis, or cholangiosepsis; (**b**) plasma IGFBP-2 levels of SIRS/sepsis patients with different causes of inflammation.

**Figure 3 biomedicines-11-03285-f003:**
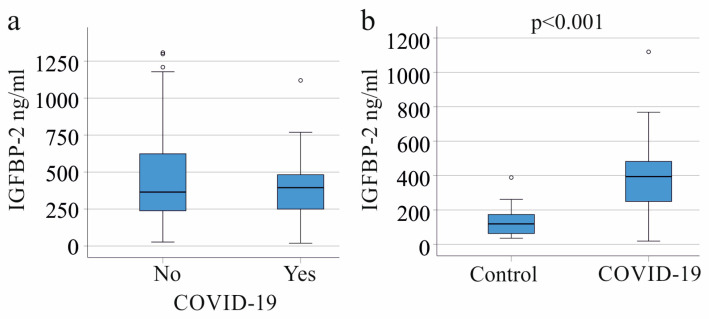
IGFBP-2 in plasma of SIRS/sepsis patients infected with SARS-CoV-2. (**a**) Plasma IGFBP-2 levels of the 24 SIRS/sepsis patients with SARS-CoV-2 infection in contrast to non-infected patients; (**b**) plasma IGFBP-2 levels of the 24 SIRS/sepsis patients with SARS-CoV-2 infection in contrast to the 22 healthy controls.

**Figure 4 biomedicines-11-03285-f004:**
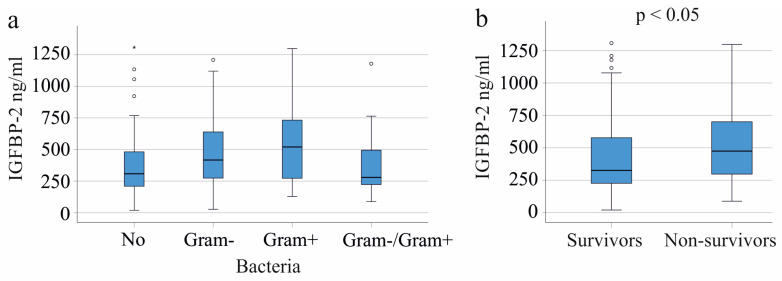
IGFBP-2 in plasma of patients with SIRS/sepsis stratified for type of bacterial infection and association of plasma IGFBP-2 with survival. (**a**) Plasma IGFBP-2 levels stratified for type of bacterial infection; (**b**) plasma IGFBP-2 levels of SIRS/sepsis survivors and non-survivors.

**Table 1 biomedicines-11-03285-t001:** Characteristics of the SIRS/sepsis patients, COVID-19 patients, and controls. Median values, and in brackets, minimum and maximum values are given. Superscript numbers are included where data from all patients are not available and indicate the number of patients who do have data available. * *p* < 0.05, *** *p* < 0.001 for comparison of the whole cohort and COVID-19 patients, ^§§§^ *p* < 0.001 for comparison of controls and patients.

Parameters	All Patients	COVID-19	Controls
Males/Females	110/47	16/8	11/11
Age (years)	59 (21–93)	63 (29–80)	58 (40–67)
Body Mass Index (kg/m^2^)	26.2 (15.4–55.6)	28.3 (20.8–45.3)	not defined
SIRS/Sepsis/Septic Shock	39/40/78 ***	0/3/21 ***	not defined
C-reactive protein mg/L	158 (12–697)	124 (44–472)	not defined
Procalcitonin ng/mL	1.16 (0.05–270.00) *	0.56 (0.08–65.40) *	not defined
Leukocytes n × 10^9^/L	10.30 (0.06–1586.00)	8.74 (2.78–18.47)	not defined
IL-6 pg/mL	89 (0–5702) ^150 §§§^	47 (6–1810) ^22 §§§^	7 (0–48) ^21 §§§^
Ferritin pg/mL	not defined	1013 (200–17,846)	not defined

**Table 2 biomedicines-11-03285-t002:** Plasma IGFBP-2 levels (ng/mL) of SIRS/sepsis patients with/without dialysis, ventilation, and, vasopressor therapy. The number of patients treated is indicated by “N” and the respective *p*-values are listed.

Intervention/Drug	SIRS/Sepsis Patients			
	N	No	Yes	*p*-Value
Dialysis	52	326 (19–1308)	458 (129–1298)	0.013
Ventilation	95	337 (27–1178)	382 (19–1308)	0.783
Vasopressor therapy	94	308 (27–1178)	406 (19–1308)	0.150

**Table 3 biomedicines-11-03285-t003:** Correlation coefficient (r) and *p*-values for the correlation of plasma IGFBP-2 with laboratory measures of inflammation.

Biomarker of Inflammation	SIRS/Sepsis		COVID-19	
	r	*p*-Value	r	*p*-Value
Leukocyte count	0.150	0.062	−0.186	0.384
Procalcitonin	0.210	0.009	0.394	0.057
C-reactive protein	0.074	0.363	−0.079	0.713
Interleukin-6	−0.070	0.396	0.216	0.334
Ferritin	not defined	not defined	0.368	0.077

## Data Availability

Data supporting reported results can be obtained from the corresponding author.

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
