# Peer review of "Plasma Insulin-like Growth Factor-Binding Protein-2 of Critically Ill Patients Is Related to Disease Severity and Survival"

_biomedicines, 2023, doi:10.3390/biomedicines11123285_

Round 1
Reviewer 1 Report
Comments and Suggestions for Authors
Dear editors:
It is a great honor and pleasure for me to be invited as the reviewer for this important work entitled “Plasma insulin-like growth factor-binding protein 2 of critically ill patients is related to disease severity and survival”. Patricia Mester1 and co-authors investigated the clinical values of Insulin-like growth factor binding-protein (IGFBP)-2 in the severely ill patients with sepsis along with their catabolic status. This study topic is novel and advanced, attributing to Prof. Christa Buechler’s long-term efforts and contributions in this scientific field. I have a number of comments concerning this study:
1. Line 136: For SIRS /sepsis diagnosis, 95% CI and p-value for ROC curve (AUROC) should be provided.
2. From the perspective of a clinician, all AUROC should be provided in all predictors for various clinical outcomes/ diagnosis to increase the readers’ interest.
The research is interesting that should be published after appropriate revision.
Comments on the Quality of English LanguageMinor editing of English language is required.
Author Response
We are very grateful to the reviewer for the constructive comments, which have helped improve the quality of our paper.
We now included IL-6 levels of controls, which were much lower in comparison to the patients (please see table 1).
- Line 136: For SIRS /sepsis diagnosis, 95% CI and p-value for ROC curve (AUROC) should be provided.
This data were added.
- From the perspective of a clinician, all AUROC should be provided in all predictors for various clinical outcomes/ diagnosis to increase the readers’ interest.
We added AUROC, p-value and 95% CI for dialysis, septic shock, COVID-19 and survival.
The research is interesting that should be published after appropriate revision.
Thank you so much for this kind comment.
Reviewer 2 Report
Comments and Suggestions for Authors
Review Biomedicines – 2760365
The authors address a very interesting topic. Research on biomarkers during sepsis or septic shock is very active.
However, the paper presents some bags.
The Plasma insulin-like growth factor-binding protein 2 it is not the only biomarker, or in any case it is already outdated.
In fact the Nephroceck also includes the Tissue Inhibitor Metalloproteinase-2 .
Both metalloproteinases represent predictive biomakers of AKI in sepsis and septic shock.
Authors are invited to read and cite very recent and very complete papers
- Acute Kidney Injury and Sepsis after Cardiac Surgery: The Roles of Tissue Inhibitor Metalloproteinase-2, Insulin-like Growth Factor Binding Protein-7, and Mid-Regional Pro-Adrenomedullin Antonio Lacquaniti , Fabrizio Ceresa , Susanna Campo , Giovanna Barbera , Daniele Caruso , Elenia Palazzo , Francesco Patanè , Paolo Monardo J Clin Med. 2023 Aug 9;12(16):5193.doi: 10.3390/jcm12165193.
The authors are invited to add in "Discussion" a paragraph on the new biomarkers and on Nephroceck in particular.
Author Response
We appreciate the constructive and insightful comments of the reviewer, which have helped improve the quality of our paper.
We now included IL-6 levels of controls, which were much lower in comparison to the patients (please see table 1).
However, the paper presents some bags.
The Plasma insulin-like growth factor-binding protein 2 it is not the only biomarker, or in any case it is already outdated.
In fact the Nephroceck also includes the Tissue Inhibitor Metalloproteinase-2 .
Both metalloproteinases represent predictive biomakers of AKI in sepsis and septic shock.
Authors are invited to read and cite very recent and very complete papers
- Acute Kidney Injury and Sepsis after Cardiac Surgery: The Roles of Tissue Inhibitor Metalloproteinase-2, Insulin-like Growth Factor Binding Protein-7, and Mid-Regional Pro-Adrenomedullin Antonio Lacquaniti , Fabrizio Ceresa , Susanna Campo , Giovanna Barbera , Daniele Caruso , Elenia Palazzo , Francesco Patanè , Paolo Monardo J Clin Med. 2023 Aug 9;12(16):5193.doi: 10.3390/jcm12165193.
The authors are invited to add in "Discussion" a paragraph on the new biomarkers and on Nephroceck in particular.
Thank you for this important update. We have included this information in Discussion. All changes are marked yellow and are easy to find.
Reviewer 3 Report
Comments and Suggestions for Authors
The authors try to demonstrate that Insulin-like growth factor binding-protein (IGFBP)-2 acts as a new biomarker for sepsis. However, the diagnosis of sepsis is not difficult so the new marker better serves as a prognostic marker for mortality or other issues.
From the data from one cohort, IGFBP-2 levels are different between survivors and non-survivals. It would be better to have a validation cohort to confirm the finding and clinical usage. Better to introduce TRIPOD guideline and multivariate analysis including other common markers.
The presentations are not standard and need redo them.
The ethical reference numbers for patients and healthy donors are required.
Comments on the Quality of English Language
None
Author Response
We are grateful to the reviewer for the valuable comments and suggestions, which have helped improve the quality of our paper. We now included IL-6 levels of controls, which were much lower in comparison to the patients (please see table 1).
The authors try to demonstrate that Insulin-like growth factor binding-protein (IGFBP)-2 acts as a new biomarker for sepsis. However, the diagnosis of sepsis is not difficult so the new marker better serves as a prognostic marker for mortality or other issues.
Diagnosis of sepsis may be still a challenge, and excellent early biomarkers are still looked for
Please see:
“Extensive research in the area is being performed to validate biomarkers, facilitate sepsis diagnosis, and allow an early intervention that, although primarily supportive, can reduce the risk of death.” https://doi.org/10.1186/s13054-021-03862-5 (published January 2022).
“The early diagnosis of sepsis is hampered by the lack of reliable laboratory measures.” ï‚· DOI: 10.1186/s12879-023-08262-4
From the data from one cohort, IGFBP-2 levels are different between survivors and non-survivals. It would be better to have a validation cohort to confirm the finding and clinical usage. Better to introduce TRIPOD guideline and multivariate analysis including other common markers.
Though IGFBP-2 was higher in non-survivors than survivors the AUROC for non-survival diagnosis was 0.618, p = 0.031 showing that this association is rather weak. Therefore, we deleted the paragraph related to multiparametric analysis because there is no good biomarker for survival included in our study.
The presentations are not standard and need redo them.
The data are shown as boxplots.
“Bar charts and box plots are omnipresent in the scientific literature. They are typically used to visualize quantities associated with a set of items. Representing the data accurately, however, requires choosing the appropriate plot according to the nature of the data and the task at hand. Bar charts are appropriate for counts, whereas box plots should be used to represent the characteristics of a distribution. https://doi.org/10.1038/nmeth.2807”
The ethical reference numbers for patients and healthy donors are required.
This information is given at the end of the journal as is requested from the journal.
“Institutional Review Board Statement: The study protocol was approved by the ethical committee of the University Hospital of Regensburg (18-1029-101) and was performed according to the updated guidelines of good clinical practice and the updated Declaration of Helsinki.”
Round 2
Reviewer 2 Report
Comments and Suggestions for Authors
The authors made the requested chianges. The paper has been significantly improved
Reviewer 3 Report
Comments and Suggestions for Authors
Much Better now